



**Mercury isotopic compositions in fine particles and offshore surface**
**seawater in a coastal area of East China: Implication for Hg sources**
**and atmospheric transformations**
Lingling Xu[a,b,*], Jiayan Shi[b,d], Yuping Chen[a,b,c], Yanru Zhang[a,b,c], Mengrong Yang[a,b],
Yanting Chen[a,b], Liqian Yin[a,b], Lei Tong[a,b], Hang Xiao[a,b], Jinsheng Chen[a,b,*]
[a] *Center for Excellence in Regional Atmospheric Environment, Institute of Urban*
*Environment, Chinese Academy of Sciences, Xiamen 361021, China*
[b] *Key Lab of Urban Environment and Health, Institute of Urban Environment, Chinese*
*Academy of Sciences, Xiamen 361021, China*
[c] *University of Chinese Academy Sciences, Beijing 100049, China*
[d] *College of Resources and Environment, Fujian Agriculture and Forestry University,*
*Fuzhou 350002, China*
* Corresponding author.
*E-mail address:* jschen@iue.ac.cn (J.S. Chen); linglingxu@iue.ac.cn (L.L. Xu)





**Abstract.** Isotopic compositions of Hg in atmospheric particles ($Hg_{PM}$) are probably
the mixed results of emission sources and atmospheric processes. Here, we present Hg
isotopic compositions in daily fine particles ($PM_{2.5}$) collected from an industrial site
(CX) and a nearby mountain site (DMS) in a coastal area of East China, as well as in
surface seawater close to the industrial area, to reveal the roles of anthropogenic
emission sources and atmospheric transformations in varying Hg isotopes. The $PM_{2.5}$
samples displayed significant spatial difference in $\delta^{202}Hg$. For the CX, the negative
$\delta^{202}Hg$ values were similar to those of source materials and $Hg_{PM}$ contents were well
correlated with other chemical tracers, indicating the dominant contributions of local
industrial activities to $Hg_{PM2.5}$. Whereas the observed positive $\delta^{202}Hg$ at the DMS was
likely associated with regional emissions and extended photo-reduction during
transport. $\Delta^{199}Hg$ values in $PM_{2.5}$ from the CX and DMS were comparable positive.
The unity slope of $\Delta^{199}Hg$ versus $\Delta^{201}Hg$ over all data suggests that the odd-MIF of
$Hg_{PM2.5}$ was primarily induced by photo-reduction in aerosols. The positive $\Delta^{200}Hg$
values with minor spatial difference were probably associated with photo-oxidation of
$Hg^0$ which is generally enhanced in the coastal environment. Total Hg in offshore
surface seawater was characterized by negative $\delta^{202}Hg$ and near-zero $\Delta^{199}Hg$ and
$\Delta^{200}Hg$, which are indistinguishable from Hg isotopes of source materials. Overall,
industrial $PM_{2.5}$ had comparable $\delta^{202}Hg$ values but more positive $\Delta^{199}Hg$ and $\Delta^{200}Hg$
as compared to surface seawater. The results indicate that atmospheric transformations
would induce significant fractionation of Hg isotopes, which obscures Hg isotopes of
anthropogenic emissions.

***Keywords:*** Particle bound mercury; Surface seawater; Hg isotopes; Mercury sources;
Photo-chemical processes.







## 1. Introduction

Mercury (Hg) is a genotoxic element and was ranked with the priority controlled pollutants in many countries. Atmospheric Hg was operationally defined as three forms: gaseous elemental mercury (GEM), gaseous oxidized mercury (GOM), and particle bound mercury (PBM or $Hg_{PM}$) (Schroeder and Munthe, 1998). Previous studies indicated that $Hg_{PM}$ concentrations in urban and industrial areas could reach up to hundreds even thousands of pg m$^{-3}$, relative to tens of pg m$^{-3}$ in uncontaminated remote areas (Fu et al., 2015; Mao et al., 2016). Hence, particulate matter (PM) can act as a vector of toxic Hg and inhalation of Hg-carrying particles is an important pathway of human exposure to atmospheric Hg. Atmospheric $Hg_{PM}$ can be directly derived from human activities and scavenged by deposition. In China, coal combustion, non-ferrous metal smelting, and cement production were considered as the three primary emission sources of atmospheric Hg (Zhang et al., 2015). On the other hand, $Hg_{PM}$ undergoes complex transformation processes in the atmosphere. $Hg_{PM}$ can be formed by the uptake of GOM in particles, which made an important contribution to $Hg_{PM}$ in heavily particle polluted areas (Xu et al., 2020). Whereas the reduction of GOM binding with dissolved organic carbon ligands in aqueous particles potentially converts it back to gas phase (Horowitz et al., 2017). The research have suggested that atmospheric $Hg_{PM}$ is generally a combined result of anthropogenic emissions and atmospheric transformations.

Analysis technique of Hg isotopes and mechanisms of Hg isotopic fractionation have come a long way in the last decade (Blum and Johnson, 2017). Hg has seven stable isotopes (including $^{196}Hg$, $^{198}Hg$, $^{199}Hg$, $^{200}Hg$, $^{201}Hg$, $^{202}Hg$, and $^{204}Hg$) and exhibits mass dependent fractionation (MDF) and mass independent fractionation (MIF) in various environmental samples and processes (Sonke and Blum, 2013; Yin et al., 2014a; Blum and Johnson, 2017). MDF of Hg isotopes is often reported as $\delta^{202}Hg$, while MIF of odd mass-numbered Hg isotopes (odd-MIF) is reported as $\Delta^{199}Hg$ and $\Delta^{201}Hg$ and MIF of even Hg isotopes (even-MIF) as $\Delta^{200}Hg$ and $\Delta^{204}Hg$. Previous laboratory and field studies have revealed that nearly all biogeochemical processes induce MDF of Hg isotopes, whereas significant odd-MIF of Hg occurs mainly in



photochemical processes (Bergquist and Blum, 2007; Malinovsky et al., 2010; Blum
et al., 2014; Sun et al., 2016a). What's more, specific ratios of $\Delta^{199}Hg/\Delta^{201}Hg$ have
been reported for different transformation processes, i.e., ~1.0 for photo-reduction and
~1.6 for photo-oxidation (Bergquist and Blum, 2007; Sun et al., 2016a). Even-MIF of
Hg isotopes is observed mostly in atmosphere related samples, which is suggested to
associate with photo-oxidation of $Hg^0$ by UV and oxidants (Chen et al., 2012; Blum
and Johnson, 2017; Fu et al., 2019). Therefore, Hg isotopes are capable of becoming
useful tracers for biogeochemical cycles of Hg in the environment.

Little literature is available on Hg isotopes of atmospheric samples due to the

difficulty in sampling enough Hg mass for isotopes analysis. Even so, Hg isotopic
compositions of atmosphere related samples, like speciated Hg, precipitation, and
lichen, have been reported in recent years (Carignan et al., 2009; Gratz et al., 2010;
Sherman et al., 2010; Chen et al., 2012; Rolison et al., 2013; Demers et al., 2013,
2015; Fu et al., 2016, 2018, 2019; Yu et al., 2016, 2020). Many studies have measured
Hg isotopes in PM to investigate its potential sources and transformation processes in
the atmosphere. In general, $Hg_{PM}$ in urban areas which is mainly impacted by local
anthropogenic sources has negative MDF and near-zero odd-MIF (Rolison et al., 2013;
Das et al., 2016; Huang et al., 2016, 2019, 2020; Huang et al., 2018; Xu et al., 2019).
While $Hg_{PM}$ in remote and coastal areas displays more significant odd-MIF, likely
linking to enhanced photo-reactions (Rolison et al., 2013; Fu et al., 2019). To date, the
fractionation of Hg isotopes in atmospheric processes has not been well elucidated,
which hampers application of Hg isotopes in tracking the transfer and transformation
paths of Hg.

This study determined Hg isotopic compositions in $PM_{2.5}$ collected from an

industrial site and a mountain site in a coastal area of East China. Comparison of
$Hg_{PM2.5}$ isotopes at the two neighbouring sites would eliminate the impacts of
meteorology and atmospheric Hg background which vary across space on $Hg_{PM}$
isotopes. Furthermore, this study measured isotopic compositions of total mercury
(THg) in surface seawater close to the industrial area and distinguished Hg isotopes
between atmospheric sample and surface media. The objective of this study is to



reveal the roles of anthropogenic sources and atmospheric transformations in varying
$Hg_{PM}$ isotopic compositions.
**2. Experiment**
*2.1. Study area description*

$PM_{2.5}$ sampling was conducted at an industrial site (Chunxiao, CX) and a nearby

mountain site (Daimeishan, DMS) on the east coast of Zhejiang province, East China
(Fig. 1). The study region experiences a typical subtropical monsoon climate, with sea
breeze in summer and continental breeze in winter. The average annual temperature,
precipitation, relative humidity, and sunshine hours were 18.1 ℃, 1608 mm, 76.8%,
and 1797 h, respectively.

The CX (121.91° E, 29.87° N, 15 m a.g.l.) is located in the Urban Environment

Observation and Research Station, Chinese Academy of Sciences, Beilun District,
Ningbo. Ningbo is a highly industrial city and there are a high density of industrial
activities around the CX. Potential Hg point sources include a large coal-fired power
plant (5000 MW) approximately 20 km to the northwest, a Chlor-alkali plant 20 km to
the northeast, and an automobile assembly plant within 1 km of the site. The CX is in
close proximity to the East China Sea (ECS, ~ 0.6 km), thus clean air masses from the
sea in warm seasons would dilute atmospheric Hg at the CX. The concentration of
GEM at the CX was reported to be 2.44 ng m$^{-3}$ from Dec. 2016 to Nov. 2017 in a
previous study (Yi et al., 2020).

The DMS (121.62° E, 29.68° N, 450 m a.s.l.) is located at the summit of

Mountain Damei, which is surrounded by trees. The site is 20 km to the coast of the
ECS and approximately 22 km south of Ningbo. There are no significant Hg point
sources within a radius of ~10 km from the DMS. However, an early study reported
that intense regional emissions and long-range transport of Hg usually caused a high
atmospheric Hg concentration at the DMS (mean: 3.3 ng m$^{-3}$, from Apr. 2011 to Apr.
2013; Yu et al., 2015).

Surface seawater samples were collected in the offshore area of Ningbo. The

seawater sampling area (about 122.04° E, 29.82° N, Fig. 1) is approximately 1 km
west of the Beilun District, Ningbo.

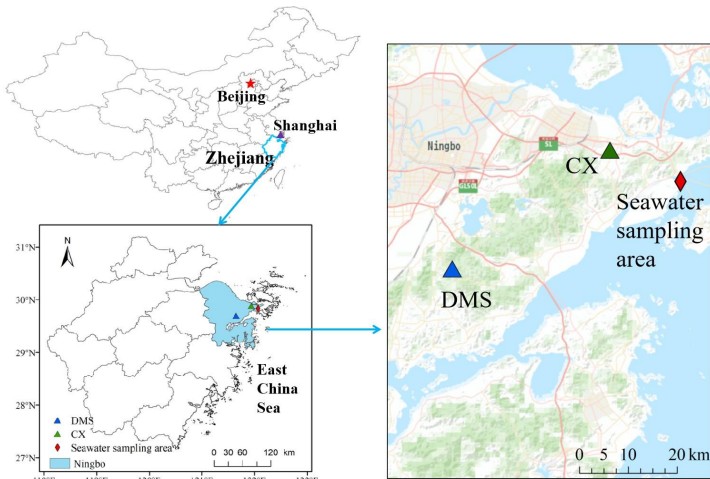

**Fig. 1** Locations of PM$_{2.5}$ (CX: industrial site; DMS: mountain site) and surface seawater sampling area.

*2.2. Sample collection and analysis*

*2.2.1. Sampling of PM$_{2.5}$*

The period of PM$_{2.5}$ sampling was from Jul. 2017 to Jun. 2018. Daily PM$_{2.5}$ samples were collected 1~2 times a week at the CX (except Jan. and Feb. 2018) and once a week at the DMS. Field blank sample was collected at each site. PM$_{2.5}$ samples were collected on preheated quartz-fiber filter (500 °C, 4 h, 8 × 10 inch, Whatman) using a high volume sampler (Tianhong TH1000H, China) with a flow rate of 1.05 m$^3$ min$^{-1}$. The filters were conditioned at 24 ± 1 °C and 52 ± 2%. The mass loading of PM$_{2.5}$ on filters was determined by mass difference before and after sampling. A total of 64 and 36 filter samples were collected at the CX and DMS, respectively. The filter samples were wrapped in aluminum foils and stored at -20 °C until analysis.

*2.2.2. Concentration of Hg and other chemical species in PM$_{2.5}$*

Six punches (ca. 0.5 cm$^2$ per punch) of each sampling filter were digested by a 10 mL of 40% aqua regia (HNO$_3$: HCl = 1:3, *v/v*) in a water bath at 95 °C for 5 min, then the solution was oxidized by 1 mL BrCl and bathed for another 30 min. After cooling down, the extracted solution was diluted to 15 mL with ultra-pure water and then analyzed by cold-vapor atomic fluorescence spectrometry (CVAFS, Brooks Rand Model III, USA) following the EPA method 1631. The content of Hg on blank filters





can be negligible (42.5 pg at the CX and 27.0 pg at the DMS) relative to those on
sample filters.
Selected $PM_{2.5}$ samples analyzed for Hg isotopes were also measured for 8 water
soluble inorganic ions ($K^+$, $Ca^{2+}$, $Na^+$, $Mg^{2+}$, $Cl^-$, $SO_4^{2-}$, $NO_3^-$, and $NH_4^+$), elemental
carbon (EC), organic carbon (OC), and levoglucosan. The water soluble ions were
analyzed by an ion chromatography (ICS-3000, Dionex, USA). EC and OC were
analyzed using a carbon analyzer (Model 4, Sunset Lab., USA) and NOISH protocol.
Analytical procedures and quality control procedures have been described by Xu et al.
(2018). Levoglucosan, an excellent indicator of biomass burning, was analyzed using
a gas chromatography – mass spectrometer detector (GC – MS, Agilent 7890A-5975C,
Agilent Tech. Inc., USA). Levoglucosan analytical procedures have been presented in
detail elsewhere (Hong et al., 2019).
*2.2.3. Sampling and analysis of Hg in seawater*
Seawater samples were collected from the surface of offshore sampling area
twice a month during Jul. 2017 ~ Jun. 2018, except Feb. 2018. Each time, three
duplicate seawater samples were collected for THg content analysis. Final THg
content was determined by the average of three duplicate samples. In addition, ~2 L
surface seawater was sampled for Hg isotopes analysis each time. The seawater
samples were stored in brown glass bottles and preserved with 1% (*v/v*) HCl in the
laboratory. They were analyzed for Hg content and isotopic compositions in a month.
Total Hg content in seawater samples was measured by the CVAFS (Brooks
Rand Model III, USA). A 25 mL of seawater sample was digested with 0.2 M BrCl at
least 12 h in advance and then analyzed using the EPA method 1631. More details can
be found in a previous study (Xu et al., 2014). Method blank was processed by bottles
filling up with ultra-pure water instead of seawater. The blank was lower than 10 pg
(n = 15), which can be negligible compared to the samples.
*2.3. Analysis of Hg isotopic compositions*
*2.3.1. $PM_{2.5}$ sample processing*
Due to effects of precipitation and short sampling duration, the mass of Hg on
most of $PM_{2.5}$ samples was not sufficient for isotopes detection. A total of 10 $PM_{2.5}$





samples from each site were analyzed for Hg isotopes. Pre-concentration of Hg from
PM$_{2.5}$ samples was conducted following a dual-stage combustion protocol (Huang et
al., 2015). To be specific, a tube furnace (OTF-1200X-II, Kejin, China) consisting of
two combustion stages was used. A sampling filter was embedded in a furnace quartz
tube (50 mm OD, 46 mm ID, 80 cm length). The tube was then placed in the furnace
so that the filter was at the first combustion stage. The second decomposition stage
was heated up in advance and maintained at 1000 °C, then the first combustion stage
was heated up to 950 °C through a temperature-programmed procedure. The released
Hg was transferred by O$_2$/Ar gas (30%/70%) at a flow rate of 20 mL min$^{-1}$ and then
trapped by a 10 mL of 40% inverse aqua regia.
*2.3.2. Seawater sample processing*
A total of 20 seawater samples were analyzed for Hg isotopes. ~2 L seawater
sample was mixed with a NH$_2$OH·HCl solution for neutralizing excess BrCl and then
a SnCl$_2$ solution for reducing the oxidized Hg. The pre-treated seawater sample was
stirred and bubbled for 1 h with Hg-free N$_2$ at a flow rate of 400 mL min$^{-1}$. The
gaseous Hg purged off seawater samples was collected by a series of three gold traps.
The gold traps were heated and the released Hg was transferred by Hg-free N$_2$ at
10~15 mL min$^{-1}$ and concentrated by a 10 mL of 40% inverse aqua regia.
*2.3.3. Hg isotopes analysis*
All trapping solutions were preserved with 1% (*v/v*) BrCl and stored at 4 °C in
the dark before Hg isotopes analysis. Hg isotopic compositions were measured by a
multi‐collector inductively coupled plasma mass spectrometer (MC-ICP-MS, Nu
Instruments Ltd. UK) following the protocols presented in a previous study (Huang et
al., 2018). Instrument mass bias was corrected using an internal standard (NIST 997
Tl) and strict sample-standard bracketing method (NIST 3133 Hg). The MDF of Hg
(represented by δ-value, ‰) is defined by the following equation (Blum and Bergquist,

2007):

$\delta^{xxx}Hg\ (‰) = [(^{xxx}Hg/^{198}Hg)sample/(^{xxx}Hg/^{198}Hg)_{NIST\ 3133} -1] \times 1000$    (1)
where xxx = 199, 200, 201, 202, and 204. The MIF of Hg (Δ-value, ‰) is calculated
using the theoretically predicted MDF as the following equation (Blum and Bergquist,



2007):

$\Delta^{xxx}Hg\ (‰) = \delta^{xxx}Hg - (\delta^{202}Hg \times \beta)$          (2)
where the mass-dependent scaling factor $\beta$ is 0.252 for $^{199}Hg$, 0.502 for $^{200}Hg$, 0.752
for $^{201}Hg$, and 1.493 for $^{204}Hg$. A reference material UM-Almaden was measured
repeatedly for quality control, yielding average $\delta^{202}Hg$ and $\Delta^{199}Hg$ values of -0.59 ±
0.10‰ (2σ, n = 25) and -0.03 ± 0.07‰ (2σ, n = 25), respectively. The results are well
consistent with those in previous studies (Blum and Bergquist, 2007; Huang et al.,
2015). The samples of this study were measured only once, so the 2σ uncertainties
derived from repeated measurements of NIST 3133 standard during each analysis
section were applied to the samples.
**3. Results and discussion**
*3.1. Concentrations and isotopic compositions of Hg$_{PM2.5}$*

Mass-based concentration of Hg$_{PM2.5}$ was used in this study to reflect reaction

processes and isotopic fractionation. The mass concentrations and isotopes of Hg$_{PM2.5}$
at industrial and mountain sites are showed in Table 1. Average mass concentrations
of Hg$_{PM2.5}$ using manual quartz filter were 0.52 ± 0.23 µg g$^{-1}$ (0.15 to 1.10, n = 51) at
the CX and 0.85 ± 0.63 µg g$^{-1}$ (0.18 to 2.80, n = 33) at the DMS, respectively. A high
Hg$_{PM2.5}$ concentration has been reported at the DMS before, which was likely due to
regional Hg emissions (Yu et al., 2015). We found that the variation coefficient (VC =
SD/Mean) of Hg$_{PM2.5}$ concentrations was lower at the CX (44.2%) than the DMS
(74.1%). In addition, the volumetric concentrations of PM$_{2.5}$ and Hg$_{PM2.5}$ were
correlated more closely at the CX ($R^2 = 0.77$, $p < 0.01$, n = 51) than the DMS ($R^2 =
0.25$, $p < 0.01$, n = 33). The data indicate that the DMS Hg$_{PM2.5}$ was influenced by
diverse sources of PM$_{2.5}$ with different Hg levels and/or complex atmospheric Hg
transformations. Spatial differences of Hg$_{PM2.5}$ were further examined by relationships
of Hg with other chemical species in PM$_{2.5}$ (Table S1). In contrast to DMS, the mass
concentrations of Hg$_{PM2.5}$ at the CX were well correlated to chemical tracers, like Cl$^-$,
NO$_3^-$, K$^+$, and OC ($r = 0.40 \sim 0.57$, $p < 0.05$, Spearson correlation), implying the
steady contributions of anthropogenic sources to Hg$_{PM2.5}$ in the industrial area.

$\delta^{202}Hg$ values for Hg$_{PM2.5}$ at the CX were in the range of -1.11‰ to 0.08‰ (mean:





-0.61 ± 0.35‰, n = 10), while $\delta^{202}$Hg values at the DMS were significantly higher and
in a larger variation from -0.78‰ to 1.10‰ (mean: 0.12 ± 0.63‰, n = 10) ($p < 0.05$, $T$
Test; Table 1 and Table S2). Hg$_{PM}$ isotopic compositions in multiple types of locations
are showed in Fig. 2 and Table S3. Negative $\delta^{202}$Hg values are generally reported for
PM in urban areas of China, such as Beijing, Changchun, Chengdu, Guiyang, and
Xi'an (Mean: from -1.60‰ to -0.42‰; Huang et al., 2015, 2016, 2019, 2020; Xu et
al., 2017, 2019; Yu et al., 2016), which are not distinguishable from those in remote
areas. Moreover, $\delta^{202}$Hg values for PM collected from urban and remote areas overlap
those from nearby anthropogenic emissions (Das et al., 2016; Yu et al., 2016; Huang
et al., 2018). In this study, the $\delta^{202}$Hg values at the CX basically fell in the variation
ranges mentioned above. Anthropogenic sources around the CX, such as industrial
factories and coal fired power plant, were most likely the main drivers of negative
MDF in PM$_{2.5}$ at this site. However, the slight positive $\delta^{202}$Hg values at the DMS have
seldom been reported in previous studies. There is no local anthropogenic emission
sources around the DMS. Thus, some additional factors may cause positive shift of
$\delta^{202}$Hg values, although the magnitudes of them producing observed $\delta^{202}$Hg are
unclear. First, backward trajectory results show that PM$_{2.5}$ samples with positive
$\delta^{202}$Hg were generally associated with air masses coming or passing through the
northeast of China (Fig. S1). An early study reported that coals in northern China have
highest $\delta^{202}$Hg value (-0.73 ± 0.33‰) compared to other regions in China (Yin et al.,
2014b). Second, adsorption of gaseous Hg$^{2+}$ on particles was suggested to be an
important contributor to Hg$_{PM}$ in the study region (Xu et al., 2020), so Hg$_{PM}$ probably
inherits significant positive MDF of Hg$^{2+}$ (Rolison et al., 2013). The results suggest
that the MDF of Hg$_{PM2.5}$ at the CX was dominantly affected by local anthropogenic
sources, while the MDF at the DMS might be a mixed result of regional emissions
and atmospheric transformations.

In contrast, $\Delta^{199}$Hg values for Hg$_{PM2.5}$ at the two sites were not different ($p > 0.05$,

$T$ Test), with comparable values of 0.17 ± 0.22‰ (from -0.17‰ to 0.52‰) at the CX
and 0.16 ± 0.24‰ (from -0.22‰ to 0.47‰) at the DMS, respectively. The $\Delta^{199}$Hg
values in this study are similar to those observed from remote areas in China (from





0.27‰ to 0.66‰; Fu et al., 2019). A laboratory study indicated that photo-reduction
of $Hg^{2+}$ restrains odd Hg in reactants (aerosols here) in priority, which shifts $\Delta^{199}Hg$
values positively (Bergquist and Blum, 2007). As shown in Table S2 and Fig. S1,
PM$_{2.5}$ samples affected by long range transport of air masses mostly had large positive
$\Delta^{199}Hg$, like PM$_{2.5}$ collected on Apr. 4, 2018 from the CX and on Jan. 10, 2018 from
the DMS. It's probably related to extensive photo-reduction of $Hg^{2+}$ in aerosols during
long range transport as previous studies suggested (Huang et al., 2016; Fu et al., 2019).
Whereas, some PM$_{2.5}$ samples affected by local air masses were also characterized by
significant positive $\Delta^{199}Hg$, like PM$_{2.5}$ collected on Apr. 4, 2018 from the CX. The
remarkable odd-MIF of Hg$_{PM}$ isotopes has commonly been reported in coastal
environment (Rolison et al., 2013; Fu et al., 2019; Yu et al., 2020), thus the positive
odd-MIF of Hg$_{PM2.5}$ in this study was likely contributed by enhanced photo-reactions.
In addition, the MIF of $^{200}Hg$, most probably relating to photo-reactions, was
significant positive and displayed no spatial difference (0.11 ± 0.07‰ at the CX and
0.14 ± 0.07‰ at the DMS; $p > 0.05$, $T$ Test), which also suggests enhanced and
homogeneous photo-reactions in the study region. It is worth noting that a part of
PM$_{2.5}$ samples collected from the DMS displayed negative $\delta^{202}Hg$ and near-zero
$\Delta^{199}Hg$, similar to those from the CX (Fig. 2). Compared with the previous study (Yu
et al., 2016), our results provide isotopes evidence that Hg$_{PM2.5}$ at the DMS was
affected by multiple sources and one of them might be regional anthropogenic
emissions.
**Table 1** Mass concentrations and isotopic compositions of Hg$_{PM2.5}$ at the industrial
site (CX) and mountain site (DMS) in East China

| Parameter [a] | CX | | DMS | |
|---|---|---|---|---|
| | Mean ± sd | Range | Mean ± sd | Range |
| Hg$_{PM2.5}$ ($\mu g\ g^{-1}$) | 0.52 ± 0.23 | 0.15 ~ 1.10 | 0.85 ± 0.63 | 0.18 ~ 2.80 |
| $\delta^{202}Hg$ (‰) | -0.61 ± 0.35 | -1.11 ~ 0.08 | 0.12 ± 0.63 | -0.78 ~ 1.10 |
| $\Delta^{199}Hg$ (‰) | 0.17 ± 0.22 | -0.17 ~ 0.52 | 0.16 ± 0.24 | -0.22 ~ 0.47 |
| $\Delta^{201}Hg$ (‰) | 0.21 ± 0.18 | -0.07 ~ 0.48 | 0.23 ± 0.36 | -0.29 ~ 0.66 |
| $\Delta^{200}Hg$ (‰) | 0.11 ± 0.07 | -0.01 ~ 0.23 | 0.14 ± 0.07 | 0.06 ~ 0.28 |
| $\Delta^{204}Hg$ (‰) | 0.19 ± 0.36 | -0.16 ~ 0.93 | 3.58 ± 3.68 | 0.26 ~ 11.38 |

[a] 51 samples collected from CX and 32 samples from DMS for Hg$_{PM2.5}$ concentration analysis; 10
samples from each site for isotope analysis.

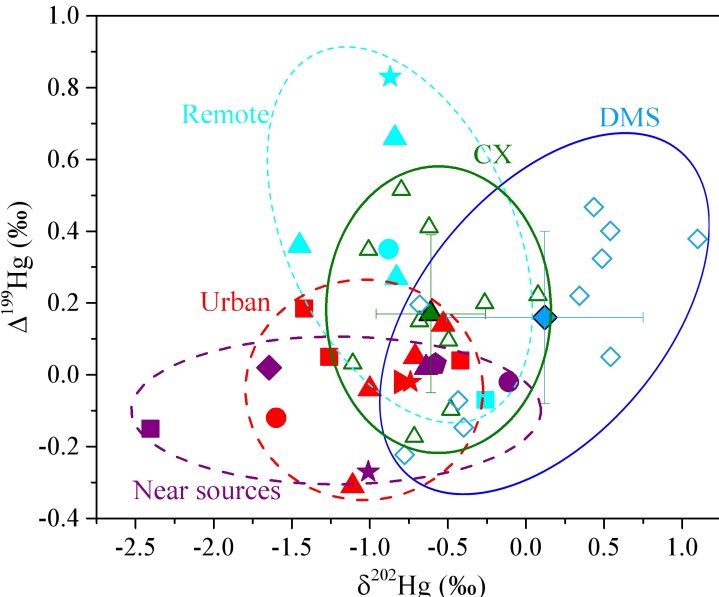


**Fig. 2**   Isotopic compositions of Hg$_{PM}$ at the multiple types of sites

(This study: ▲△ mean and each values at the CX, ◆◇ mean and each values at the DMS;
Remote sites: ★coast, ■▲mountain, ●island (Rolison et al., 2013; Yu et al., 2016; Fu et al.,
2019); Urban sites in China: ▲Beijing, ●Changchun, ★Chengdu, ■Guiyang, ▶Xi'an (Huang
et al., 2015, 2016, 2019, 2020; Xu et al., 2017, 2019; Yu et al., 2016); Sites near emission sources:
▲■industrial, ●volcano, ●landfill, ◆traffic, ★near CFPP (Das et al., 2016; Huang et al., 2018;
Yu et al., 2016; Zambardi et al., 2009)

### *3.2. Influence of anthropogenic emissions on MDF of Hg$_{PM2.5}$*

Prior studies have compiled Hg isotopic compositions of major source materials,
such as fossil fuels, non-ferrous metal ores, and crustal rocks, and they generally
display large negative $\delta^{202}$Hg and negative or near-zero $\Delta^{199}$Hg values (Huang et al.,
2016; Sun et al., 2016). Combustion or/and industrial processing induces limited MIF
(Sun et al., 2013; Sun et al., 2016), so we assumed that emitted Hg conserves odd-Hg
isotopes of source materials. The $\Delta^{199}$Hg values for most Hg$_{PM2.5}$ are distinguishable
from those of source materials, indicating that anthropogenic emissions were not the
drive factors for odd-MIF of Hg$_{PM2.5}$ in the study region. As for MDF, above analyses
indicated that the MDF of Hg$_{PM2.5}$ at the CX is subjected to local anthropogenic
sources, while the MDF at the DMS is probably caused by the combination of
atmospheric transformations and regional emissions. Spearson correlation between





$\delta^{202}Hg$ and chemical components was conducted to explore the impacts of
anthropogenic emissions on $Hg_{PM2.5}$. The study region is highly industrialized, thus
industrial emissions are likely important contributors to $Hg_{PM2.5}$ isotopes. It's a pity
that we did not measure metallic elements to trace industrial contributions. As shown
in Table S4, the $\delta^{202}Hg$ values were associated with $Cl^-$, $SO_4^{2-}$, and levoglucosan
contents in PM2.5, which are generally considered as indicatives of coal combustion
and biomass burning.

### *(1) Coal combustion*

Coal combustion was considered to be the primary Hg emission source in China,

which accounted for 47.2% of total anthropogenic emissions (253.8 t out of 537.8 t).
Coal combustion was also the dominant Hg emission source in Zhejiang province,
with a contribution of ~50% to total Hg emissions (Zhang et al., 2015). Hg isotopic
compositions of coals in China have large variations in MDF with $\delta^{202}Hg$ values from
-2.36‰ to -0.14‰ (Biswas et al., 2008; Yin et al., 2014b). A prior study reported that
emitted $Hg_{PM}$ has a shift of -0.5‰ relative to $\delta^{202}Hg$ of coal feeds (Sun et al., 2014).
Hence $\delta^{202}Hg$ values for $Hg_{PM}$ emitted from coal combustion in China were estimated
to be -2.86‰ to -0.64‰. The $\delta^{202}Hg$ values at the CX basically overlap and slightly
shift to positive, while the $\delta^{202}Hg$ values at the DMS have a large positive shift as
compared to those for emitted $Hg_{PM}$ from coal combustion.

In this study, $Cl^-$ was mainly originated from coal combustion, given that $Cl^-$

content in PM2.5 was not correlated to $Na^+$. Besides, $SO_4^{2-}$ was primarily transformed
from $SO_2$ which is mainly emitted from coal combustion. The $\delta^{202}Hg$ values at the CX
were significantly correlated to $Cl^-$ content ($R^2 = 0.46$, $P < 0.05$, Fig. 3a) and well
associated with $SO_4^{2-}$ content in PM2.5 ($R^2 = 0.38$, $P = 0.056$, Fig. 3b). The results
imply that coal combustion played an important role in the MDF of $Hg_{PM2.5}$ at the CX.
It should be noted that there are many metal smelting factories near the CX. We did
not measure the tracers for smelting, but a previous study reported mean $\delta^{202}Hg$ value
for non-ferrous metal ores as $-0.47 \pm 0.77$‰ (Yin et al., 2016). We assumed that Hg
emitted from non-ferrous metal smelting conserves the isotopes of source materials
due to lack of data for processing at current stage (Sun et al., 2016). Then, less
negative MDF of Hg from non-ferrous metal smelting could explain the positive-shift
MDF at the CX relative to coal combustion emissions. It is reasonably inferred that
the MDF of Hg$_{PM2.5}$ at the CX is a result of multiple anthropogenic sources such as
coal combustion and non-ferrous metal smelting. Differently from the CX, the $\delta^{202}$Hg
values at the DMS were significantly correlated to SO$_4^{2-}$ ($R^2 = 0.68$, $P < 0.05$, Fig. 3b),
but not to Cl$^-$ ($P > 0.05$). It seems unlikely that coal combustion was the predominant
contributor to the positive MDF at the DMS. Whereas under the influence of transport,
the transformation of SO$_2$ to SO$_4^{2-}$ usually enhances and the photo-reduction of Hg$^{2+}$
in aerosols tends to extensive which would shift $\delta^{202}$Hg to positive to a certain extent
(Bergquist and Blum, 2007). The results imply that coal combustion emissions in a
regional scale or from long range transport had a potential impact on the MDF of
Hg$_{PM2.5}$ at the DMS, which is consistent with an earlier study conducted at the same
site based on Hg concentration and trajectory analysis (Yu et al., 2015).
*(2) Biomass burning*
Total Hg emissions from biomass burning were estimated to be 3 ~ 4 t (Zhang et
al., 2015), which is very small relative to coal combustion. Whereas during some
times, like autumn harvesting and spring wildfire occurring seasons, biomass burning
could become a major contributor to atmospheric Hg (Giglio et al., 2013; Huang et al.,
2016; Fu et al., 2018). Previous studies have reported that biological materials display
negative $\delta^{202}$Hg and $\Delta^{199}$Hg values, like foliage ($\delta^{202}$Hg: -2.67‰ to -1.79‰; $\Delta^{199}$Hg:
-0.47‰ to -0.06‰), litterfall samples ($\delta^{202}$Hg: -3.03‰ to -2.35‰; $\Delta^{199}$Hg: -0.44‰ to
-0.22‰), and lichen ($\delta^{202}$Hg: -2.32‰ to -1.83‰; $\Delta^{199}$Hg: -0.35‰ to -0.22‰) (Demers
et al., 2013; Jiskra et al., 2015; Yin et al., 2013; Yu et al., 2016; Zheng et al., 2016).
Such negative $\delta^{202}$Hg and $\Delta^{199}$Hg of biological materials can not explain the isotopes
of Hg$_{PM2.5}$ in this study. Moreover, the contribution of biomass burning is supposed to
shift $\Delta^{199}$Hg values negative, but we found no significant negative correlation between
$\Delta^{199}$Hg and K$^+$ or levoglucosan as indicative of biomass burning influence from the
whole study period (Table S4). The results suggest that biomass burning was not the
dominant contributor to Hg$_{PM2.5}$ in the study region.
Interestingly, we found a close negative correlation between $\delta^{202}$Hg and





levoglucosan content in PM$_{2.5}$ at the CX ($R^2 = 0.67$, $P < 0.05$, Fig. 3c) excluding a
PM$_{2.5}$ sample collected on Dec. 19, 2017. Considering no relevant study on Hg
isotopic fractionation during burning processes so far, it can be assumed that Hg
emitted from biomass burning conserves large negative MDF signature of biological
materials. Thus, we cannot rule out the possibility that the contribution of biomass
burning led to a negative deviation of δ$^{202}$Hg values at the CX to some extent.
Actually, the contribution of biomass burning to Hg$_{PM2.5}$ is often substantial in a short
time period (i.e., Mar. 2018, Fig. S2a, *https://firms.modaps.eosdis.nasa.gov/*), which
can explain a weak correlation between Δ$^{199}$Hg and K$^+$ or levoglucosan in PM$_{2.5}$. In
this study, the most negative odd-MIF was observed for PM$_{2.5}$ samples collected on
Mar. 21, 2018, with Δ$^{199}$Hg value of -0.17‰ at the CX and -0.22‰ at the DMS. The
finding was likely related to biomass burning, since those PM$_{2.5}$ samples were
associated with air masses originating from or passing through the northeast of China
with dense fire spots (Fig. S2b).


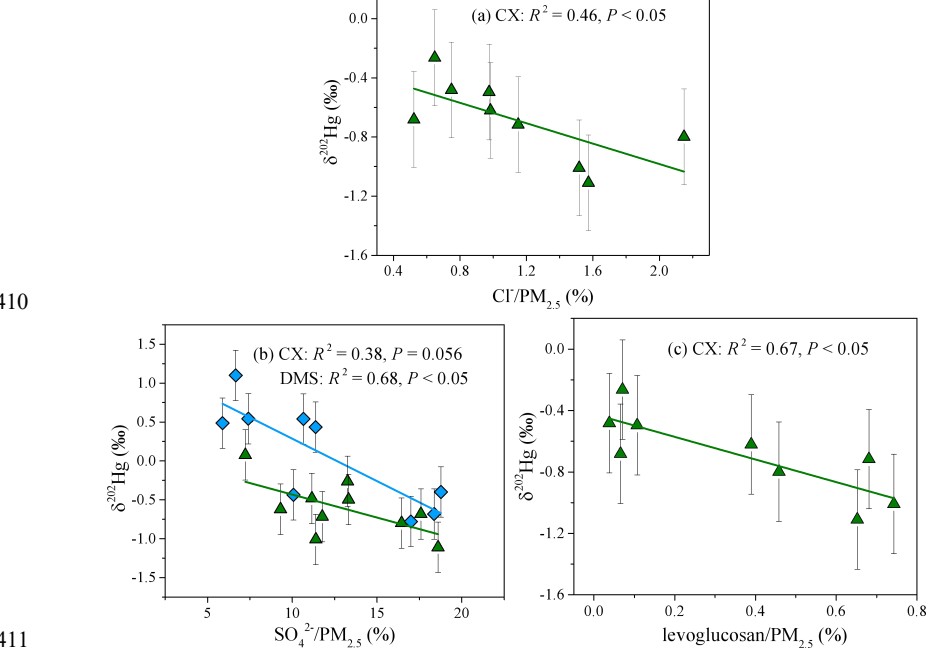

**Fig. 3**    Relationships of δ$^{202}$Hg with (a) Cl$^-$, (b) SO$_4^{2-}$, and (c) levoglucosan contents
in PM$_{2.5}$ at CX (▲) or DMS (◆). Uncertainty (2σ) for δ$^{202}$Hg in PM$_{2.5}$ is 0.25‰.
***3.3. Influence of photo-chemical process on isotopes of Hg$_{PM2.5}$***





Large odd-MIF of Hg isotopes in most $PM_{2.5}$ samples of this study was likely
related to photo-chemical processes. An experiment study has found that the oxidation
of $Hg^0$ by halogen atom (Cl· or Br·) results in a negative shift of $\Delta^{199}Hg$ in product
$Hg^{2+}$ (Sun et al., 2016). Given that partitioning of $Hg^{2+}$ between gas and particle
phases leads to limited odd-MIF of Hg isotopes (Wiederhold et al., 2010, Fu et al.,
2019), then the formation of $Hg_{PM}$ via oxidation of $Hg^0$ and following adsorption on
particles could not explain the positive odd-MIF of $Hg_{PM2.5}$ in this study. Previous
experiments and field studies have reported that photo-reduction of $Hg^{2+}$ in aqueous
solution induces the odd-MIF of Hg isotopes and results in large positive $\Delta^{199}Hg$
values (Bergquist and Blum, 2007; Zheng and Hintelmann, 2009, 2010). Hence,
photo-reduction of inorganic $Hg^{2+}$ in aerosols could be proposed to be a key factor for
the odd-MIF of $Hg_{PM2.5}$ in the study region. The linear relationship between $\Delta^{199}Hg$
and $\Delta^{201}Hg$ is often used to identify MIF processes of odd-Hg isotopes. The slope of
$\Delta^{199}Hg$ versus $\Delta^{201}Hg$ yielded from overall data was 0.92 in this study ($R^2 = 0.83$, $P <$
0.01; Fig. 4a). The near-unity slope of $\Delta^{199}Hg$ versus $\Delta^{201}Hg$ was widely observed in
particles from other studies (Rolison et al., 2013; Huang et al., 2016, 2019; Fu et al.,
2019; Xu et al., 2019). The $\Delta^{199}Hg/\Delta^{201}Hg$ ratio of this study is consistent with the
indicative ratio of aqueous photo-reduction of inorganic $Hg^{2+}$ (~1.0, Bergquist and
Blum, 2007; Zheng and Hintelmann, 2009), but different from the ratios of other
processes, like photo-oxidation (1.64 by Br· and 1.89 by Cl·, Sun et al., 2016) and
photo-demethylation (1.36, Bergquist and Blum, 2007). Isotopic compositions of
$Hg_{PM}$ are usually the combined effects of many environmental processes, like above
mentioned photo-reactions and various anthropogenic sources. Thus, the observed
odd-MIF of $Hg_{PM2.5}$ in the study region seems like a "net" result of aqueous
photo-reduction process.
The similarity of odd-MIF anomaly between the CX and DMS suggests the
photo-reduction of $Hg^{2+}$ in aerosols was homogeneous on a regional scale. However,
relationships of $\Delta^{199}Hg$ with $Hg_{PM2.5}$ content and $\delta^{202}Hg$ showed distinct spatial
difference. For the DMS, the $\Delta^{199}Hg$ values generally decreased with $Hg_{PM2.5}$ content
increased (Fig. 4b) and the correlation between $\Delta^{199}Hg$ and $\delta^{202}Hg$ was significantly



positive ($R^2 = 0.56$, $P < 0.05$; Fig. 4c). Experimental studies have indicated that the
photo-reduction of $Hg^{2+}$ releases $Hg^0$ and preferentially retains odd and heavier
isotopes in solutions (Bergquist and Blum, 2007; Zheng and Hintelmann, 2009),
which is expected to result in a positive relationship between $\Delta^{199}Hg$ and $\delta^{202}Hg$ and
an inverse relationship between $\Delta^{199}Hg$ and $Hg_{PM2.5}$ content. In this study, the
consistent relationships of $\Delta^{199}Hg$ with $\delta^{202}Hg$ and $Hg_{PM2.5}$ at the DMS strongly imply
a predominant role of photo-reduction in isotopic fractionation of $Hg_{PM2.5}$ at this site.
In addition, $\delta^{202}Hg$ signatures of anthropogenic emissions from regional and
long-range transport might be largely obscured by photo-reduction process, which
well explains the positive $\delta^{202}Hg$ at the DMS. In contrast, the variation of $\Delta^{199}Hg$ at
the CX was not associated with $Hg_{PM2.5}$ contents or $\delta^{202}Hg$. The result suggests an
insignificant impact of photo-reduction relative to anthropogenic sources on MDF and
Hg content in $PM_{2.5}$ at the CX. On the other hand, a better coincidence of high $Hg_{PM2.5}$
and high $\delta^{202}Hg$ (Fig. 4d) supports that constant local Hg emissions dominantly
affected $Hg_{PM2.5}$ content at the CX.

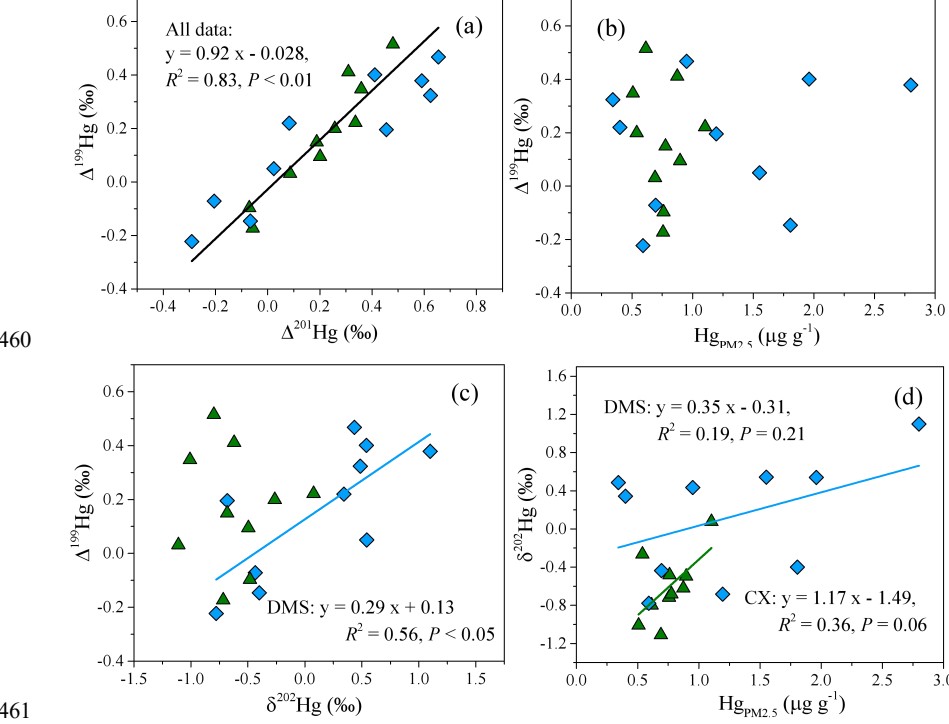






**Fig. 4**   Linear relationships between (a) $\Delta^{199}Hg$ and $\Delta^{201}Hg$, (b) $\Delta^{199}Hg$ and $Hg_{PM2.5}$
content, (c) $\Delta^{199}Hg$ and $\delta^{202}Hg$, and (d) $\delta^{202}Hg$ and $Hg_{PM2.5}$ content at the CX ( ▲ ) and
DMS ( ◆ ). Uncertainty ($2\sigma$) for $\Delta^{199}Hg$ and $\delta^{202}Hg$ in $PM_{2.5}$ is 0.03‰ and 0.25‰,
respectively.
***3.4. Potential mechanism of even-MIF***

A small but significant MIF of $^{200}Hg$ was observed in most $PM_{2.5}$ samples from

this study, with mean $\Delta^{200}Hg$ values of 0.11 ± 0.07‰ at the CX and 0.14 ± 0.07‰ at
the DMS. They are more positive than those in urban (mean = 0.01‰ to 0.09‰, Das
et al., 2016; Huang et al., 2016; Xu et al., 2017) and remote areas (mean = 0.07‰ to
0.10‰, Fu et al., 2019), but similar to those in coastal and island areas (Rolison et al.,
2013; Fu et al., 2019). In general, Hg emitted from anthropogenic sources has $\Delta^{200}Hg$
of near-zero (Sun et al., 2016b), while large $\Delta^{200}Hg$ values are mainly observed in
atmospheric samples, i.e., precipitation, gaseous $Hg^{2+}$, and aerosols (Chen et al., 2012;
Rolison et al., 2013; Fu et al., 2019). Significant even-MIF of Hg isotopes has been
suggested to associate with photo-oxidation of $Hg^0$, from upper troposphere or/and
from in situ involving UV light and oxidants (Chen et al., 2012; Fu et al., 2019). This
could help explain significant $\Delta^{200}Hg$ values in coastal areas where halogen atoms are
expected to be abundant. The $\Delta^{200}Hg$ values in $PM_{2.5}$ were not different between sites,
similar to $\Delta^{199}Hg$ values, which supports that the observed $\Delta^{200}Hg$ were associated
with photo-chemical processes of minor spatial difference.

An experimental study showed that gas-phase oxidation of $Hg^0$ vapor by Cl and

Br atoms results in positive $\Delta^{200}Hg$ and large negative $\Delta^{199}Hg$ values in products (Sun
et al., 2016a). So direct oxidation of $Hg^0$ in particles could explain positive $\Delta^{200}Hg$ but
positive $\Delta^{199}Hg$ values in this study. Based on previous field and experimental studies,
we can preferably hypothesize several phases during $Hg_{PM}$ transformations. (1)
gas-phase oxidation of $Hg^0$. This process generally enhances in areas with abundant
halogen atoms (Wang et al., 2019), which would result in detectable positive $\Delta^{200}Hg$
values in products (gaseous $Hg^{2+}$) (Sun et al., 2016a). (2) gas-particle partitioning of
$Hg^{2+}$. This process made an important contribution to $Hg_{PM}$ in the study region (Xu et
al., 2020), but it is strongly temperature-dependent that unlikely produces the MIF of
Hg isotopes (Fu et al., 2019). (3) aqueous photo-reduction of $Hg^{2+}$ in particles. This



process induces positive odd-MIF of Hg isotopes as previously discussed. The
proposed speculation can basically explain for $\Delta^{199}$Hg and $\Delta^{200}$Hg values in coastal
and island areas, although there are still some uncertainties, like fractionation of Hg
isotopes during gas-particle partitioning.
The MIF of $^{204}$Hg has only been reported in a few studies, presenting small
positive values for TGM and negative values in precipitation and remote PM samples
(Demers et al., 2013, 2015; Fu et al., 2019). Interesting, we observed a distinct spatial
difference in $\Delta^{204}$Hg values, with 3.58 ± 3.68‰ (from 0.26‰ to 11.38‰) at the DMS
and 0.19 ± 0.36‰ at the CX (lower than 2SD of repeated NIST 3177 analysis). The
large positive $\Delta^{204}$Hg at the DMS have not been reported in atmospheric samples
before. In addition, there was no correlation between $\Delta^{204}$Hg and $\Delta^{200}$Hg in this study,
which is not consistent with the early finding that $\Delta^{204}$Hg values were complementary
to $\Delta^{200}$Hg (Demers et al., 2013; Fu et al., 2019). It's generally speculated that
even-MIF of Hg isotopes is derived from photo-oxidation of $Hg^0$ to $Hg^{2+}$ (Chen et al.,
2012). However, the difference in spatial distribution for $\Delta^{200}$Hg and $\Delta^{204}$Hg in this
study not very support this speculation. Up to now, the mechanisms for even-MIF of
Hg isotopes remain unknown and we can not give a further explanation for $\Delta^{204}$Hg in
$PM_{2.5}$ in this study.

### 3.5. Isotopes of Hg in adjacent surface seawater

Hg isotopes are often used to track the transport and transformations of Hg in the
environment. The average concentration of THg in seawater was 10.5 ± 5.0 ng $L^{-1}$,
with a range of 1.9 ~ 23.6 ng $L^{-1}$ (Table S1). As shown in Fig. 5, the concentrations of
seawater THg displayed distinct time variations, with higher levels during Sep. ~ Mar.
than during Apr. ~ Aug, which is likely related to precipitation cycle. The average
$\delta^{202}$Hg value of seawater samples was -1.31 ± 0.59‰, with most samples fell in the
range of -2.00‰ ~ -1.00‰. Whereas Hg-MIF in seawater samples was not significant,
with mean $\Delta^{199}$Hg, $\Delta^{201}$Hg, and $\Delta^{200}$Hg values of -0.02 ± 0.07‰, 0.00 ± 0.05‰, and
0.04 ± 0.03‰, respectively. The negative MDF and near-zero MIF of surface seawater
are well consistent with those of source materials (Huang et al., 2016; Sun et al.,
2016b), suggesting the dominant effect of anthropogenic emissions on Hg in offshore





surface seawater. A minor change in intensity of industrial activities as expected
among the months also supports the above deduction.

Isotopic compositions of THg in surface seawater and Hg$_{PM}$ at the adjacent

industrial site are consistent in MDF but not in MIF. Similar results were obtained
when comparing to wet deposition which presents negative $\delta^{202}$Hg and positive
$\Delta^{199}$Hg and $\Delta^{200}$Hg (Chen et al., 2012; Huang et al., 2018). Negative MDF of Hg in
industrial PM$_{2.5}$ and adjacent surface seawater implies an important role of local
anthropogenic sources. On the other hand, the unity slope of $\Delta^{199}$Hg versus $\Delta^{201}$Hg
($\Delta^{199}$Hg = 1.12 × $\Delta^{201}$Hg - 0.02, $R^2$ = 0.68, n = 19) indicates that the odd-MIF of Hg
isotopes in surface seawater was mainly produced by photo-reduction of Hg$^{2+}$.
Whereas, the minor $\Delta^{199}$Hg anomalies suggest that photo-reduction was not evident
for surface seawater. A big discrepancy in the MIF of Hg isotopes between
atmospheric samples and surface seawater further evidences that atmospheric
transformations would induce significant MIF of Hg isotopes, which obscures Hg
isotopic signatures of anthropogenic emissions.

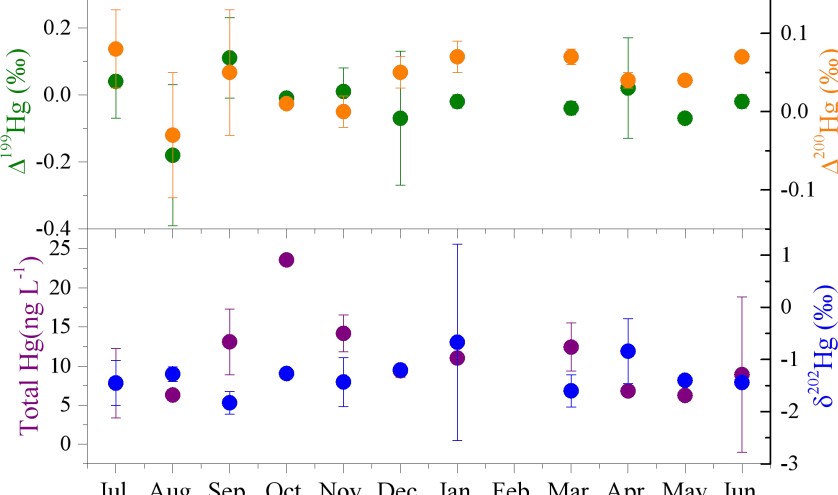

**Fig. 5**    Monthly variations of total Hg concentration, $\delta^{202}$Hg, $\Delta^{199}$Hg and $\Delta^{200}$Hg of

surface seawater during the sampling period from July 2017 to June 2018

**4. Conclusion**

This study investigated Hg isotopic compositions in PM$_{2.5}$ collected from nearby

industrial and mountain sites in a coastal area and also in adjacent offshore surface





seawater. $Hg_{PM2.5}$ displayed significant spatial difference in MDF but not in odd-MIF.
Negative $\delta^{202}Hg$ in $PM_{2.5}$ at the CX was primarily induced by local industrial
activities like coal combustion. Whereas, the slight positive $\delta^{202}Hg$ at the DMS could
not be fully explained by anthropogenic emissions. Other than the effect of regional
transport, a close correlation between $\delta^{202}Hg$ and $\Delta^{199}Hg$ at the DMS implies that
photo-chemical processes likely contributed to the MDF of $Hg_{PM2.5}$. Significant
positive odd-MIF of $Hg_{PM2.5}$ and the unity slope of $\Delta^{199}Hg$ versus $\Delta^{201}Hg$ indicate an
important role of photo-reduction in aerosols. The observed positive $\Delta^{200}Hg$ values in
this study were probably associated with photo-oxidation of $Hg^0$ which is generally
enhanced in the coastal environment. THg in surface seawater was characterized by
negative MDF and near-zero odd-MIF, which is more consistent with isotopic
signatures of source materials. The anomalies of Hg-MIF were larger for atmospheric
$PM_{2.5}$ than for surface seawater, suggesting that atmospheric transformations induce
significant MIF of Hg isotopes and obscure Hg isotopic signatures of initial emissions.
This study illustrates that the comparison of Hg isotopic compositions among relevant
media is more effective to identify Hg emission sources and atmospheric
transformations.

**Novelty statement**
Comparison of isotopic compositions of $Hg_{PM2.5}$ was conducted between nearby
industrial and mountain sites, which is more effective to reveal the roles of
anthropogenic emission sources and transformation processes in varying Hg isotopes.
Hg isotopic compositions in industrial $PM_{2.5}$ and in offshore surface seawater were
also compared. The results indicate that atmospheric transformations would induce
significant fractionation of Hg isotopes and obscure the specific Hg isotopic
signatures of initial emissions.

**Data availability.** HYSPLIT trajectory model and gridded meteorological data
(Global Data Assimilation System, GDAS1) are available from the US National
Oceanic and Atmospheric Administration (http://ready.arl.noaa.gov). Fire data are



available in the Fire Information for Resource Management System (FIRMS,
https://firms2.modaps.eosdis.nasa.gov/map/#d:2021-04-26..2021-04-27;@6.7,2.0,3z).
All data in this study are available upon request to the first author via email
(linglingxu@iue.ac.cn).

**Author contributions**. JSC, LLX, and YRZ designed this study. MRL, LQY, YTC,
LT and HX conducted the sampling. YRZ and LLX participated in sample treatment
and measurements. LLX wrote the paper. JYS and YPC helped the graphics
production. All authors reviewed the paper.

**Competing interests.** The authors declare that they have no conflict of interest.

**Acknowledgements.** This research was financially supported by National Natural
Science Foundation of China (No. 21507127; 41575146 & U1405235), Natural
Science Foundation of Fujian province (2016J05050), the Cultivating Project of
Strategic Priority Research Program of Chinese Academy of Sciences (XDPB1903),
the CAS Center for Excellence in Regional Atmospheric Environment (E0L1B20201),
and Xiamen Atmospheric Environment Observation and Research Station of Fujian
Province.

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
