# Peer review of "Mercury isotopic compositions in fine particles and offshore surface seawater in a coastal area of East China: Implication for Hg sources and atmospheric transformations"

_Atmospheric Chemistry and Physics, 2021_

## Author Response (AR2)

Response to Editors and Reviewers

-Editor

Response: We appreciate for the opportunity to revise the paper. We thank the two reviewers for the comments which are helpful for us to improve the paper. We have addressed the comments and revised the manuscript accordingly. We hope the revised manuscript could meet the quality of ACP.

Comment on acp-2021-493

Anonymous Referee #1

The Manuscript entitled 'Mercury isotopic compositions in fine particles and offshore surface seawater in a coastal area of East China: Implication for Hg sources and atmospheric transformations' investigated the Hg isotopic composition of fine aerosols (PM2.5) sampled from industrial and mountain sites in a coastal area if East China. In addition, the authors also evaluated the Hg isotopes in surface seawater close to the Industrial area. The authors aimed to obtain the roles of anthropogenic sources and atmospheric transformations in particulate Hg isotopic compositions. Stable Hg isotopes have become a useful proxy for the identification of Hg sources, particularly as a result of improvements in high-precision analytical methods. Limited data are available on the stable isotopes of Hg or their application in source apportionment in atmospheric aerosols. Therefore, studies on atmospheric Hg and its isotopic compositions are important for understanding the atmospheric concentrations, sources, transport mechanisms, and fate of particulate Hg and the data are important to the broad scientific community. The manuscript is well written and the results are discussed in detail, although, some of the latest studies are not reviewed. Hence, I suggest the acceptance of this manuscript in ACP after minor suggestions below are addressed.

**Response:** We appreciate for your overall positive evaluation of the manuscript. We have revised the manuscript carefully according to the suggestions. The "point to point" responses are as follows and the main correlations are marked in red in a PDF file "Revised manuscript with changes marked".

A little more on atmospheric particulate mercury (PBM) and it scenario (literature review) is need in the introduction section.

The motivation to carry out this study must me made clear with more gaps identified.

**Response:** As you suggested, we have indicated the role of $Hg_{PM}$ in the cycling of Hg in the manuscript (lines 79-82). In addition, we made a literature review of $Hg_{PM}$ isotopes and elaborated the motivation of this study more clearly (lines 105-121 and 128-132). The main revisions are as follows.

"In addition, $Hg_{PM}$ has a residence time of several weeks as it can transport and deposit at a regional scale (Selin, 2009). The research has suggested that atmospheric $Hg_{PM}$ is generally a combined result of anthropogenic emissions and atmospheric processes, which plays a crucial role in the global cycling of Hg"

"East China is densely populated and one of the heaviest industrialized area in China. The concentration of Hg$_{PM}$ in this region has been well characterized (Hong et al., 2016; Xu et al., 2020;Yu et al., 2015), but only two studies conducted at the remote sites have referred to Hg$_{PM}$ isotopes (Fu et al., 2019;Yu et al., 2016). To the best of our knowledge, there is no report on the isotopic compositions of Hg$_{PM}$ from urban areas of East China. Likewise, the effect of atmospheric processes on the fractionation of Hg isotopes in the coastal region has not been well elucidated."

"The objectives of this study are (1) to differentiate the Hg isotopes in PM$_{2.5}$ from the two neighboring industrial and mountain sites; (2) use the Hg isotopes to explore the influence of anthropogenic sources on the Hg$_{PM}$; (3) to reveal the role of atmospheric transformations in varying Hg$_{PM}$ isotopic compositions."

Line 98-112: The literature review missed some of the recent works on PBM isotopic ratios of atmospheric samples (e.g., Source identification of atmospheric particle-bound mercury in the Himalayan foothills through non-isotopic and isotope analyses; Atmospheric particle-bound mercury in the northern Indo-Gangetic Plain region: Insights into sources from mercury isotope analysis and influencing factors).

**Response:** We have introduced the recent works on the application of Hg$_{PM}$ isotopes in sources or transboundary Hg transport identification (e.g., Fu et al., 2019EST; Guo et al., 2021EP, 2022GF, lines 108-115) in the manuscript.

Line 246-249: The authors presented the Hg mass in PM2.5, however I did not find the PBM concentrations presented and discussed. The Hg mass can also suggest the source is from natural or anthropogenic. When assessing Hg enrichment and sources, the PBM/PM ratio may be useful if we have Hg concentrations for natural and anthropogenic components (e.g., soil and coal) in the region of interest? Please check it for the two studied sites.

**Response:** We have added the discussion of Hg$_{PM}$ volumetric concentration in the manuscript (lines 272-277). In addition, we agree that PBM/PM ratio (i.e. Hg mass concentration) could indicate that the source is from natural or anthropogenic. We did not find the Hg mass for natural and anthropogenic components in the study region, so we addressed this issue based on the national data (lines 283-296). We found that the Hg contents of PM$_{2.5}$ in the study region are higher than those of natural sources (e.g., dust and topsoil; 0.056 ~ 0.30 μg g$^{-1}$; Schleicher et al., 2015) and those of coals in China (mean: 0.22 μg g$^{-1}$; Yin et al., 2014b).

Line 258: Spearson correlation? Should be Spearman?

**Response:** Sorry for the typo. The "Spearson" should be "Spearman".

The Hg isotope data presented here does not seem to be able to distinguish between different sources. For example, Hg isotopes (Figure 2) show urban, remote and near sources, however, the clear sources e.g., coal, industrial emission, traffic and soils are all possible source of particulate Hg? This is not clear and not discussed clearly. Distinguishing between these sources seems difficult based on isotope alone. Thus I am not sure why the authors conclude anthropogenic sources (what are the sources) is not clear.

**Response:** We agree that we could not identify the specific sources of Hg$_{PM2.5}$ solely based on Hg isotopes, because the $\delta^{202}$Hg values of potential sources are not distinguishable. We have revised the content and clarified this point as follows (lines 305-313).

"The $\delta^{202}$Hg values at the CX basically overlap those for PM in urban areas of China (mean: from -1.60‰ to -0.42‰), as well as those for major source materials such as coal combustion, smelting, and cement plants (mean: -1.10‰, -0.87‰, and -1.42‰ respectively, Huang et al., 2016) and those for PM near anthropogenic emissions such as industry, landfill, traffic, and coal-fired power plants (mean: from -2.41‰ to -0.58‰) (Fig. 2). The result likely indicates an important contribution of anthropogenic sources to the CX Hg$_{PM2.5}$. However, the $\delta^{202}$Hg values of above mentioned potential sources are not distinguishable, thus we could not identify the specific sources of Hg$_{PM2.5}$ solely based on Hg isotopes."

Line 343: Why the authors directly start with numbering 1. Coal combustion, this may break the flow and so on?

**Response:** Thank you for the suggestion. We have revised the section 3.2 to make the text more concise and fluent.

Similarities or differences in Hg isotope ratios at the two sites need to be described and the different seasons of their collection reported. The authors should see if their results plotted on a coherent mixing line on an inverse Hg concentration plot (i.e. d202Hg vs 1/HgP). Soils and values for PM from other locations in China might also be informative on such a plot. More broadly, Hg isotope ratios in aerosols from coastal sites should be compared with those in aerosols from other locations in Asia. This may be placed in Supplementary document.

**Response:** (1) As you suggested, we have presented the ratios of $\Delta^{199}$Hg to $\Delta^{201}$Hg at the both sites and compared them with the ratios in aerosols from coastal site and from other locations in Asia in the manuscript (lines 449-454) as follows. On the other hand, we did not present the Hg isotopes ratios among seasons, because the number of the samples in each season was not large enough.

"The slope of $\Delta^{199}$Hg versus $\Delta^{201}$Hg yielded from the data of each site was 1.16 ($R^2 = 0.92$) at the CX and 0.63 ($R^2 = 0.85$) at the DMS, respectively. The data over the two sites defined a straight line with a slope of 0.92 ($R^2 = 0.83$, $P < 0.01$; Fig. 4a). The near-unity slope of $\Delta^{199}$Hg versus $\Delta^{201}$Hg was widely observed in particles from coastal site and from other locations in Asia (Fu et al., 2019; Rolison et al., 2013; Huang et al., 2016, 2019; Xu et al., 2019). The $\Delta^{199}$Hg/$\Delta^{201}$Hg ratios of this study are more consistent with the indicative ratio of aqueous photo-reduction of inorganic Hg$^{2+}$ (~1.0, Bergquist and Blum, 2007; Zheng and Hintelmann, 2009), but different from the ratios of other processes, like photo-oxidation (1.64 by Br· and 1.89 by Cl·, Sun et al., 2016) and photo-demethylation (1.36, Bergquist and Blum, 2007)."

(2) We have plotted a line on $\delta^{202}$Hg vs. Hg$_{PM}$ concentration in Fig. 3a and the relevant discussion was showed as follows (lines 369-371). As you suggested, we have plotted the relationship of $\delta^{202}$Hg with 1/Hg$_{PM}$ for this study and for other locations in China and Asia. The relationships of $\delta^{202}$Hg with 1/Hg$_{PM}$ were similar to those with Hg$_{PM}$ concentration. The linear relationship was basically insignificant for the DMS, the CX and over the total data. For above reasons, we did not discuss the relationships of $\delta^{202}$Hg with 1/Hg$_{PM}$ further.

"The result was supported by the correlation between $\delta^{202}$Hg values and $Hg_{PM2.5}$ concentrations which was insignificant at the DMS, but significant at a loose level at the CX (Fig. 3a)."

Plot of $\Delta$199Hg (‰) vs. $\delta$202Hg (‰) is not presented. Hg-MIF ($\Delta$199Hg) signatures are also valuable for distinguishing Hg contamination pathways because Hg2+ photo-reduction in aerosols. The authors discussed on the slope, however, it is important to show the figure to clearly understand the atmospheric transformation and photochemical process.

**Response:** The plot of $\Delta^{199}$Hg vs. $\delta^{202}$Hg has already been presented in Fig. 4c in the submitted manuscript. We agree that Hg-MIF ($\Delta^{199}$Hg) signatures are valuable for distinguishing Hg contamination pathways and the relevant discussion is presented in the section 3.1 (lines 323-332) as follows. In addition to the slope of $\Delta^{199}$Hg vs. $\delta^{202}$Hg, we also presented the relationships of $\Delta^{199}$Hg with $\delta^{202}$Hg and Hg content to reveal the role of photo-reduction in aerosols (Fig. 4bc, lines 463-478). We found an inverse relationship between $\Delta^{199}$Hg and $Hg_{PM2.5}$ content and a positive correlation between $\Delta^{199}$Hg and $\delta^{202}$Hg at the DMS, which suggest a key role of photo-reduction of $Hg^{2+}$ in isotopic fractionation of $Hg_{PM2.5}$. In contrast, the variation of $\Delta^{199}$Hg at the CX was not associated with $Hg_{PM2.5}$ contents or $\delta^{202}$Hg. The result suggests an insignificant impact of photo-reduction relative to anthropogenic sources on MDF and Hg content in $PM_{2.5}$ at the CX.

"The significant positive $\Delta^{199}$Hg in this study are similar to those observed in coastal areas (Rolison et al., 2013; Yu et al., 2020) and in remote areas in China (Fu et al., 2019), but distinguishable from those in urban and industrial areas with near-zero values due to anthropogenic emissions (Das et al., 2016; Huang et al., 2016, 2018, 2020; Xu et al., 2019; Yu et al., 2016). A laboratory study has indicated that photo-reduction of $Hg^{2+}$ restrains odd Hg in reactants (aerosols here) in priority, which shifts $\Delta^{199}$Hg values positively (Bergquist and Blum, 2007). Thus, it's reasonably supposed that the positive odd-MIF of $Hg_{PM}$ in the study region was associated with photo-reduction of $Hg^{2+}$ in aerosols."

Line 421-424: This statement needs more thought. Photo-reduction of Hg2+ mostly results in positive D199Hg in reactant Hg.

**Response:** The reactant here is aerosols. To clarify it, we have revised the sentence.

"A laboratory study has indicated that photo-reduction of $Hg^{2+}$ restrains odd Hg in reactants (aerosols here) in priority, which shifts $\Delta^{199}$Hg values positively (Bergquist and Blum, 2007)."

Line 530-532: Please show in figure as suggested previously.

**Response:** As you suggested, we presented the plot of $\Delta^{199}$Hg vs. $\Delta^{201}$Hg as Fig. S3 in the supplementary document.

The detailed revisions are needed before publications.

**Response:** We have checked and revised the whole manuscript carefully before re-submission.

Comment on acp-2021-493

Anonymous Referee #2

The proposed paper described Hg isotope variation of PM$_{2.5}$ sample collected from urban and mountain area of East China to test Hg isotope as the tracer of source and process of particulate Hg in atmosphere. Since Hg isotope of particulate Hg is still scarce, the data provided by this study surely contribute to better understanding of Hg chemistry in the atmosphere. The authors well cover the previous monitoring and experimental studies, and they tried to interpret their data set through comparing the relevant works. Nevertheless, two points have to be considered to evaluate this work correctly. Firstly, description of methodology section is insufficient. As authors mentioned, the technical difficult is accurate measurement of trace amount of particulate Hg in PM$_{2.5}$ sample. I cannot validate quality of the data only from the provided information in methodology section (see specific comments). Secondly, missing of Hg0 data makes all interpretation rather speculative. Gaseous elemental Hg is the predominant form of Hg in atmosphere, while gaseous oxidizing Hg and particulate Hg (likely contribution of Hg(II) is high) occupy minor pool. Conversion of Hg species from large to minor pool potentially causes large isotope fractionation. I think authors should mention the isotopic variation of GEM in China more carefully to interpret their data. The specific comments are as below.

**Response:** We appreciate for your valuable comments and suggestions which helped us to improve the quality of the article. According to your comments, we have given more information about sample treatments and measurements in the methodology section, and considered the isotopes of GEM when interpreting the MIF of Hg isotopes in PM$_{2.5}$. The specific responses to the comments are as follows and the main correlations are marked in red in a PDF file "Revised manuscript with changes marked".

L66. Despite HgPM level expressed here being volume based, their own results are expressed as mass basis. It makes comparison difficult.

**Response:** In addition to volume based Hg$_{PM}$ level, we have compared the contribution of Hg$_{PM}$ to total Hg between industrial and uncontaminated areas in the manuscript (lines 64-69).

"Previous studies indicated that Hg$_{PM}$ concentrations in urban and industrial areas could reach up to hundreds even thousands of pg m$^{-3}$, relative to tens of pg m$^{-3}$ in uncontaminated remote areas (Fu et al., 2015; Mao et al., 2016). In addition, Hg$_{PM}$ can account for up to 40% of atmospheric Hg in industrial areas, relative to < 5% in uncontaminated areas (Guo et al., 2022;Schroeder and Munthe, 1998)."

L92. ~1.0 for photo-reduction of Hg(II); L93. ~1.6 for photo-oxidation of Hg(0)

**Response:** We have revised the sentence as "…~1.0 for photo-reduction of Hg$^{2+}$ and ~1.6 for photo-oxidation of Hg(0)".

L144. What is "regional emission"? It should be specified.

**Response:** The "regional emissions" are mainly industrial activities and coal combustion in the Yangtze River Delta and the neighboring region of Anhui, Jiangsu, and Zhejiang Provinces (Yu et al., 2015). We have specified the regional emissions in the manuscript (lines 152-155).

L147. Although I thought seawater data is rather minor focus in this paper, more oceanographic background should be provided to help data interpretation, such as temperature and primary productivity.

**Response:** Thank you for the suggestion. We have presented the salinity and pH of the seawater samples in the manuscript (lines 158-160).

"The salinity of the seawater samples ranged from 21.2‰ to 29.5‰. The pH of the seawater samples was in the range of 5.7 ~ 8.5, with the mean value of 7.5 ±0.6."

L200. This means, authors pooled 10 samples to be one? If so, it should be written accordingly.

**Response:** Nope. Individual $PM_{2.5}$ samples with sufficient Hg mass were chosen for Hg isotopes analysis. There were 10 samples in total for each site.

L202~. Recovery through this combustion process should be given at the last of this paragraph. Careful operation is often required for complete recovery using dual combustion furnace.

**Response:** Thank you for the suggestion. We have presented a detail operation and given the Hg recovery of the dual-stage protocol accordingly (lines 220-229).

"The combustion procedure was run with no samples in the furnace quartz tube before $PM_{2.5}$ sample treatment every day to remove residual volatiles. The released Hg was transferred by $O_2/Ar$ gas (30%/70%) at a flow rate of 20 mL $min^{-1}$ and then trapped by a 10 mL of 40% inverse aqua regia (2: 4: 9 ratio of 10 M HCl, 15 M $HNO_3$ and ultra-pure water) in a designed glass bottle. In advance of $PM_{2.5}$ sample analysis, the accuracy of dual-stage combustion method was assessed by the analysis of the standard NIST SRM 3133 Hg (dripped on blank filters) and the certified reference material GBW07434. The Hg recovery efficiency of the dual-stage protocol was in the range of 87.6% ~ 103.3% (mean: 95.0 ±5.1%, n = 6)."

L214 Concentration of SnCl2 should be given.

**Response:** We have given the concentration of $SnCl_2$ (200 g/L) in the manuscript.

L220-227. The description of MC-ICP-MS analysis is poor although they cited one reference paper. The method here is CV-MC-ICP-MS? If so, it should be noted. In which aqueous Hg concentration did author choose to the isotope analysis? Did author match the Hg signal of sample and standard? The UM-Almaden values were obtained by exactly same level to the sample? Since sample measurements were made only one time, the information are important to validate data quality.

**Response:** Yes, the method here is CV-MC-ICP-MS. The pre-concentration solutions were measured in Xiamen University (Xiamen, China) with the method described in a recently published paper (Huang et al., 2021). According to the suggestion, we have given more information of sample measurements (lines 241-253) and data quality assurance (lines 261-265) in the manuscript as follows. In addition, after careful consideration, we have deleted $^{204}Hg$ data due to its low natural abundance.

"Hg isotopic compositions were measured by a multi–collector inductively coupled plasma mass spectrometer (MC-ICP-MS, Nu Instruments Ltd. UK) equipped with an introduction device

following the protocols presented in previous studies (Huang et al., 2015; Huang et al., 2021;Lin et al., 2015). The introduction device includes a modified cold-vapor generator (CVG) and an Aridus III nebulizer for respective Hg and Tl introduction. Between standard and sample, the CVG was rinsed with 3% (v/v) $HNO_3$ solution to ensure the Hg signal returned to the background level. Instrument mass bias was corrected using both an internal standard (NIST 997 Tl) and a strict sample-standard bracketing method (NIST 3133 Hg). A reference material NIST 8610 was measured repeatedly for quality control. The pre-concentration solutions were diluted to about 1.5 ~ 3.0 ng mL$^{-1}$ and the NIST 3133 and NIST 8610 were kept at 2.0 ng mL$^{-1}$ during the analysis period."

"The repeated measurement of NIST 8610 during the analysis session yielded $\delta^{202}$Hg and $\Delta^{199}$Hg values of -0.60 ± 0.15‰ and -0.02 ± 0.06‰ (2σ, n = 7). In addition, a well-known reference material UM-Almaden showed a long-term average of $\delta^{202}$Hg = -0.59 ± 0.10‰ and $\Delta^{199}$Hg = -0.03 ± 0.07‰ (2σ, n = 25), which are well consistent with those in previous studies (Blum and Bergquist, 2007; Huang et al., 2015)."

L246. Again, why the author showed only mass-based concentration. Besides, there are no data of total mass of particle on the filter. Without this value, readers cannot calculate concentration of Hg in final solution used for the isotope analysis. If the author used hydride generation system such as CETAC HGX-100, >2.5 ng is required for the precise data analysis.

**Response:** We showed only mass-based concentration because the mass-based concentration is more appropriate for discussion of reaction processes and isotope fractionation. As you suggested, we have presented the mass and the volumetric concentrations of Hg$_{PM2.5}$, and the concentration of PM$_{2.5}$ in the manuscript (Table 1). We used a modified CVG for Hg introduction in the isotope analysis. All of the pre-concentration solutions were diluted to approximately 1.0~3.0 ng mL$^{-1}$ (at least 1.0 ng mL$^{-1}$). The internal precision of $\delta^{202}$Hg for each measurement was determined as about 0.035‰ ~ 0.055‰ corresponding to the concentration of 1.0~3.0 ng mL$^{-1}$ (Lin et al., 2015).

L249. Again, what is the regional emission?

**Response:** As you suggested, we have specified the regional emissions (lines 281-283) in the manuscript.

L300. remarkably positive odd-MIF

**Response:** We have revised the related sentence.

L302. L299-302. I couldn't understand the reasoning here. What is the enhanced photo□reaction? Hg0 reduction? Or MMHg demethylation? The δ202Hg vs Δ199Hg of DMS in Fig. 2 seems positively correlated with slope being ca. 0.4. Does this trend support author's interpretation?

**Response:** We did not state this question clearly. The significant positive $\Delta^{199}$Hg and the near-unity slope of $\Delta^{199}$Hg vs. $\Delta^{201}$Hg in the study region indicate that odd-MIF in PM$_{2.5}$ was impacted by photo-reduction of Hg$^{2+}$. The correlation of $\delta^{202}$Hg and $\Delta^{199}$Hg at the DMS (Fig. 4c) was consistent with the experimental results of photo-reduction that generally showed positive correlation for the residual Hg pool (here aerosols). The result supports the interpretation that photo-reduction of Hg$^{2+}$ was the important source of the odd-MIF of Hg$_{PM}$ at the DMS. This issue was discussed in more detail in Section 3.3 (lines 464-475).

Here, we have revised the content as follows focusing on the comparison of $\Delta^{199}$Hg values (lines 324-332).

"The significant positive $\Delta^{199}$Hg in this study are similar to those observed in coastal areas (Rolison et al., 2013; Yu et al., 2020) and in remote areas in China (Fu et al., 2019), but distinguishable from those in urban and industrial areas with near-zero values due to anthropogenic emissions (Das et al., 2016; Huang et al., 2016, 2018, 2020; Xu et al., 2019; Yu et al., 2016). A laboratory study has indicated that photo-reduction of $Hg^{2+}$ restrains odd Hg in reactants (aerosols here) in priority, which shifts $\Delta^{199}$Hg values positively (Bergquist and Blum, 2007). Thus, it's reasonably supposed that the positive odd-MIF of HgPM in the study region was associated with photo-reduction of $Hg^{2+}$ in aerosols."

L349. A prior study estimated that…of coal feeds based on the mass balance model (Sun et al., 2014).

**Response:** We have revised the sentence as "A prior study estimated that emitted $Hg_{PM}$ has a shift of -0.5‰ relative to $\delta^{202}$Hg of coal feeds based on the mass balance model (Sun et al., 2014)."

Figure 4. I am afraid poor data quality from the rather scattered correlation of Δ199Hg and Δ201Hg.

**Response:** Thank you for the suggestion. We have compared the correlation coefficient of $\Delta^{199}$Hg and $\Delta^{201}$Hg of this study with other public studies. We found that the correlation of $\Delta^{199}$Hg and $\Delta^{201}$Hg of this study was comparable with those conducted on Chinese urban areas ($r^2 = 0.81\sim0.92$ in Beijing, Huang et al., 2019; $r^2 = 0.92$ in Beijing, 0.73 in Changchun, and 0.76 in Chengdu, Xu et al., 2019), and better than the study conducted during three cruises to Chinese seas ($p > 0.05$, Yu et al., 2020). Thus, we thought that the rather scattered correlation of $\Delta^{199}$Hg and $\Delta^{201}$Hg was probably because the Hg contents in environmental samples are low and they are affected by complex factors. In addition, according to the above suggestions, we have given more information about sample measurements and the results of quality control in the methodology section to validate the data quality.